# Does cardiorespiratory fitness mediate or moderate the association between mid-life physical activity frequency and cognitive function? findings from the 1958 British birth cohort study

**Tom Norris** [ID], **John J. Mitchell, Joanna M. Blodgett, Mark Hamer, Snehal M. Pinto Pereira** [ID]*

Faculty of Medical Sciences, Institute of Sport, Division of Surgery and Interventional Science, Exercise and Health, UCL, London, United Kingdom

* snehal.pereira@ucl.ac.uk

## Abstract

### Background

Physical activity (PA) is associated with a lower risk of cognitive decline and all-cause dementia in later life. Pathways underpinning this association are unclear but may involve either mediation and/or moderation by cardiorespiratory fitness (CRF).

### Methods

Data on PA frequency (exposure) at 42y, non-exercise testing CRF (NETCRF, mediator/moderator) at 45y and overall cognitive function (outcome) at 50y were obtained from 9,385 participants (50.8% female) in the 1958 British birth cohort study. We used a four-way decomposition approach to examine the relative contributions of mediation and moderation by NETCRF on the association between PA frequency at 42y and overall cognitive function at 50y.

### Results

In males, the estimated overall effect of 42y PA $\geq$ *once per week (vs. <once per week*) was a 0.08 (95% confidence interval: 0.04,0.13) higher overall cognitive function z-score at 50y. The estimated controlled direct effect was similar (0.08 (0.03, 0.12)). Thus, the proportion of the estimated effect via mediation or moderation by NETCRF was small (~3%), with confidence intervals straddling the null. In females, there was no estimated overall effect of PA frequency on overall cognitive function.

### Conclusion

We present the first evidence from a four-way decomposition analysis of the potential contribution that CRF plays in the relationship between mid-life PA frequency and subsequent

**Data Availability Statement:** The original data for the 1958 NCDS are available from the UK Data Service; applications for access to any data held by

the UK Data Archive that forms part of the NCDS Biomedical Resource will require special license and should be submitted to clsfeedback@ioe.ac.uk.

**Funding:** This work was funded by a UK Medical Research Council Career Development Award (ref: MR/P020372/1) and Senior Non-clinical fellowship (ref: MR/Y009398/1) awarded to SPP. MH and JB are supported by a British Heart Foundation grant (SP/F/20/150002). MH is also supported by the NIHR University College London Hospitals Biomedical Research Centre. JJM is funded by an MRC grant (MR/N013867/1). The funders had no role in study design, data collection and analysis, decision to publish, or preparation of the manuscript. There was no additional external funding received for this study.

**Competing interests:** The authors have declared that no competing interests exist

cognitive function. Our lack of evidence in support of CRF mediating or moderating the PA frequency—cognitive function association suggests that other pathways underpin this association.

## Introduction

Cognitive impairment, especially in later-life, is concerning, because it raises the risk of developing various dementias [1]. Among older people, dementia is a major cause of disability. For example, it accounts for almost 12% of years lived with disability due to non-communicable diseases [2]. As a result of population ageing, identifying strategies to alleviate cognitive impairment (and therefore also dementia) is critical [2]. While Lecanemab has very recently been approved in the US and Japan for the treatment of Alzheimer's disease, in the United Kingdom (UK), there remains no effective treatments to reverse or delay dementia progression [3]. Thus, identifying modifiable causal factors which can be intervened on to delay age-related declines in cognitive function is essential.

Physical activity (PA) is one such modifiable factor. Higher levels of PA are associated with lower risk of all-cause dementia [4,5], cognitive decline [6] and better later-life cognition [7]. However, pathways underpinning the PA—cognition association are unclear. While cardiovascular [8] and mental [5] health may be involved, an additional pathway may include cardiorespiratory fitness (CRF). CRF refers to the capacity of the circulatory and respiratory systems to supply oxygen, and the ability of muscle tissue, to utilise it [9]. PA, particularly aerobic activity, has established beneficial effects on CRF [10] and evidence suggests that higher CRF is associated with greater subsequent cognitive function [11]. Thus, CRF may have a mediating role in the PA—cognition association. However, the link between PA and CRF is likely to be dynamic, because, for example, those with higher levels of fitness might participate in PA more often and/or at greater volumes/intensity. Thus, in addition to CRF potentially mediating the PA—cognition association, CRF may also moderate this association, with the association differing depending on fitness levels. However, evidence is mixed and may be specific to certain cognitive domains. For example, some studies suggest that the effect of PA on memory is stronger in fitter individuals [12], whilst others find the positive effect of PA on processing speed is stronger in less-fit individuals [13].

It is plausible that CRF exerts both mediating and moderating effects on the PA—cognition association. Conventional analyses that ignore the potential mediation and moderation of PA by CRF on cognitive outcomes can therefore be biased [14]. Thus, the ability to incorporate both these effects into analyses is critical for obtaining a more accurate insight into underlying mechanisms through which PA influences cognition. To our knowledge, no study has investigated the mechanisms underlying the PA—cognition association considering potential mediation and moderation by CRF simultaneously. Therefore, we attempt to address this critical knowledge gap using a four-way decomposition approach [15] in an age-homogenous general population sample born in 1958 and followed-up throughout their lives. Our objective was to formally examine the relative contribution of mediation and moderation by CRF on the PA—cognition association.

## Materials and methods

Data come from the 1958 British birth cohort [16], which enrolled 17,638 participants at birth during a single week in March 1958 in Great Britain and subsequently added a further 920 immigrants born in the same week. In mid-adulthood (45 years (y)), participants remain

broadly representative of the original study sample [17]. At 42y (n = 11,419) and 50y
(n = 9,790) participants completed face-face interviewer-administered questionnaires in their
homes. At 45y (n = 9,377) they were visited by a nurse who collected biomedical, physical and
sociodemographic information.

Our analytic sample (N = 9,385) includes participants with valid measures on all cognitive tests
undertaken at 50y (see S1 Fig). Ethical approval was given by the London multi-centre research
ethics committee for the age 42y and 50y (08/H0718/29) sweeps, whilst the South-East multi-cen-
tre research ethics committee (01/1/44) provided ethical approval for the age 45y sweep. Written
informed consent was obtained from participants at all ages. Full details are available elsewhere
[16,18]. Data were accessed for research purposes on 27/8/21. Authors had no access to informa-
tion that could have identified individual participants during or after data collection.

## Exposure: Physical activity frequency (42y)

In 2000, when participants were 42y, they were asked whether, and how often they participated
in a range of leisure-time physical activities (e.g., competitive sports of any kind, keep-fit clas-
ses, running; further details in S1 Text). We derived a dichotomous variable representing
whether people participated in leisure-time physical activities '*less than once per week*' (0) or
'*at least once per week*' (1).

## Mediator: Non-exercise testing cardiorespiratory fitness (NETCRF, 45y)

In 2003 when participants were 45y (range 44y-46y), weight and height were measured using
standard protocols [19]; body mass index (BMI; kg/m$^2$) was calculated. After a few minutes
rest, three measurements of resting heart rate (RHR, beats/minute) were obtained (taken one
minute apart) using an automated device (Omron 907 blood pressure monitor, Omron
Healthcare, UK); mean RHR was calculated. A modified version of the European Prospective
Investigation into Cancer and Nutrition Physical Activity Questionnaire (EPIC-PAQ) was
used to assess leisure-time PA [20].

NETCRF was predicted using collected information on sex, age, and 45y data on BMI,
RHR, and self-reported PA. Of the three NETCRF algorithms originally developed [21], we
selected the algorithm developed [22], and previously used [23], in UK populations. This algo-
rithm has shown a multiple correlation of 0.76 against exercise testing estimated CRF. It also
has a high level of cross-validity (0.71–0.80) when applied in samples other than the one it was
developed in [21].

NETCRF at 45y was calculated as [23]:

$$\text{NETCRF}_{45y} \text{ (mL/kg/min)} = [\text{sex coefficient} \times 2.78 - (\text{age}_{45y} \times 0.11) - (\text{BMI}_{45y} \times 0.17) - (\text{RHR}_{45y} \times 0.05) + (\text{PA}_{45y} \text{ level coefficient}) + 21.41]$$

Where, the sex coefficient was 1 for males and 0 for females. The PA coefficients were based
on adherence to contemporaneous guidelines assessed using EPIC-PAQ, and mapped onto
the prediction equation, similar to previous studies [23]: 0.0 for inactive during leisure-time;
0.29 for active, but not meeting guidelines; and, 1.21 for meeting guidelines (i.e., at least
150minutes/week moderate-intensity or 75minutes/week vigorous-intensity PA). NETCRF
estimates were converted into maximal aerobic capacity metabolic equivalent (MET) values (1
MET corresponds to an oxygen consumption of 3.5 mL/kg/min (based on a 40y male, weigh-
ing 70kg) [23]).

## Outcome: Overall cognitive function (50y)

'Overall cognitive function' was calculated as the average score of four cognitive function tasks
completed in 2008 when participants were 50y [24]. The four tasks, completed in the following

order, were: i) immediate verbal memory; ii) verbal fluency; iii) visual processing speed; and iv) delayed verbal memory. These tests have been routinely used as measures of cognition in large-scale epidemiological studies [25] and are predictive of incident dementia [26]. For immediate verbal memory, participants were played an audio recording of 10 words and were given two minutes to orally recall them. Verbal fluency was assessed via an animal naming test, in which respondents were given one minute to name as many animals as they could think of. Visual processing speed was assessed using a dual-letter cancellation test, in which participants were presented with blocks of letters and were asked to read through the blocks from left to right, crossing out 'Ws' and 'Ps' as fast as they could. Search speed was calculated by summing the total number of letters scanned, including both target and non-target letters. Delayed recall was measured by asking participants to recall as many words as they could from the original list presented to them during the first word-recall task, with a two-minute cut-off. This delayed memory recall task was done approximately five minutes after the initial recall task. The four measures were standardised (z-score, separately for each sex) and the average (overall cognitive function) was calculated.

## Potential confounders

Potential confounders of the PA-CRF-cognitive function association were identified a-priori and included in the directed acyclic graph (S2 Fig). Baseline confounders included: Social class at birth, childhood cognitive function (11y), sports participation (16y), educational attainment (33y), smoking status (42y) and alcohol consumption (42y). Intermediate confounders (i.e., confounders of the CRF (45y) cognitive function (50y) association that could be affected by physical activity (42y)) included BMI (45y), PA level (45y), sleep problems (45y) and self-rated health over the previous 12 months (45y); details in Supplementary materials. Of note, the *causal* diagram in S2 Fig, aims to illustrate underlying causal networks underpinning the relationship between 42y PA and 50y cognitive function. This is distinct from the *prediction equation* for NETCRF at 45y described above. As such, variables that perform well in prediction models may not be causal [27]. Similarly, some variables e.g. BMI, can perform well in *predicting*, and also, *cause* CRF.

## Statistical analysis

Due to observed sex differences in the PA−cognition association [28], and the CRF distribution [29], we chose a-priori to perform sex-stratified analyses.

Regression models were used to explore the relationship between i) PA at 42y and cognitive function at 50y; ii) PA at 42y and NETCRF at 45y; and iii) NETCRF at 45y and cognitive function at 50y. Models were first unadjusted and then adjusted for baseline confounders listed above (the NETCRF−cognition association was additionally adjusted for PA at 42y, and intermediate confounders measured at 45y: BMI, PA level, sleep problems and self-rated health over the previous 12 months). Importantly, for the PA at 42y and cognitive function at 50y association, we did *not* adjust for NETCRF or the intermediate confounders. This is because adjusting for these intermediate variables would attenuate the overall effect, which is the estimand of interest here. To address missingness in data presented in Table 1, we used multiple imputation by chained equations following recommended guidelines [30]. We assume the data are missing-at-random and strengthen this assumption by including in the imputation models all variables described above (i.e., 42y PA, 45y NETCRF, 50y cognitive function and potential confounders), as well as childhood internalizing and externalizing behaviours and cognitive ability which have been shown to predict missingness in follow-up [17]. The number of imputations (n = 30) required to achieve convergence of parameter estimates was

**Table 1. Sample characteristics (n = 9,385).**

| Variable (reporting age (y)) | Males (n = 4,614) | | Females (n = 4,771) | |
|---|---|---|---|---|
| | Missing (n (%)) | N(%) / Mean (SD) | Missing (n (%)) | N(%) / Mean (SD) |
| *Exposure*: Physical activity (42y) | 358 (7.8) | | 262 (5.5) | |
| *<once/week* | | 1,434 (33.7) | | 1,533 (34.0) |
| *At least once/week* | | 2,822 (66.3) | | 2,796 (66.0) |
| *Mediator*: NETCRF (METS) (45y) | 1,683 (36.5) | 11.8 (1.2) | 1856 (38.9) | 9.0 (1.3) |
| *Outcome*: Overall cognitive function (z-score) (50y) | - | 0.0 (0.6) | - | 0.0 (0.7) |
| *Covariates* | | | | |
| Social class (birth)[†] | 723 (15.7) | | 734 (15.4) | |
| *Professional/managerial* | | 783 (20.1) | | 774 (19.2) |
| *Skilled non-manual* | | 393 (10.1) | | 396 (9.8) |
| *Skilled manual* | | 1,836 (47.2) | | 1,942 (48.1) |
| *Partly skilled* | | 461 (11.9) | | 480 (11.9) |
| *Unskilled/Other* | | 418 (10.7) | | 445 (11.0) |
| Childhood cognition score (11y) | 610 (13.2) | 17.5 (7.8) | 630 (13.2) | 17.2 (7.3) |
| Sports participation (16y) | 1,264 (27.4) | | 1,266 (26.5) | |
| *No chance* | | 54 (1.6) | | 159 (4.5) |
| *Hardly ever* | | 413 (12.3) | | 1,203 (34.3) |
| *Sometimes* | | 1,075 (32.1) | | 1,372 (39.1) |
| *Often* | | 1,808 (54.0) | | 771 (22.0) |
| Educational attainment (33y) | 740 (16.0) | | 604 (12.7) | |
| *<O-levels* | | 1,333 (34.4) | | 1,209 (29.0) |
| *<Undergraduate degree* | | 1,928 (49.8) | | 2,447 (58.7) |
| *≥ Degree* | | 613 (15.8) | | 511 (12.3) |
| Smoking status (42y) | 357 (7.7) | | 260 (5.5) | |
| *Never* | | 1,933 (45.4) | | 2,142 (47.5) |
| *Ex-smoker* | | 1,152 (27.1) | | 1,107 (24.5) |
| *Current smoker* | | 1,172 (27.5) | | 1,262 (28.0) |
| Alcohol consumption frequency (42y) | 356 (7.7) | | 260 (5.5) | |
| *Never* | | 150 (3.5) | | 257 (5.7) |
| *Rarely* | | 338 (7.9) | | 772 (17.1) |
| *2–4 times/month* | | 1,158 (27.2) | | 1,466 (32.5) |
| *≥2 times/week* | | 2,612 (61.3) | | 2,016 (44.7) |
| Self-rated health (45y) | 821 (17.8) | | 827 (17.3) | |
| *Poor* | | 59 (1.6) | | 70 (1.8) |
| *Fair* | | 630 (16.6) | | 632 (16.0) |
| *Good* | | 2,417 (63.7) | | 2,556 (64.8) |
| *Excellent* | | 687 (18.1) | | 686 (17.4) |
| Sleep score (45y)*[α] | 754 (16.4) | 0 (0, 1) | 763 (16.0) | 0 (0, 2) |
| Physical activity (45y) | 1640 (35.5) | | 1,795 (37.6) | |
| *Inactive* | | 934 (31.4) | | 982 (33.0) |
| *Active-not meeting guidelines* | | 358 (12.0) | | 502 (16.9) |
| *Active-meeting guidelines* | | 1682 (56.6) | | 1,492 (50.1) |
| BMI (kg/m$^2$) (45y) [α] | 783 (17.0) | 27.3 (25.0, 30.1) | 810 (17.0) | 25.7 (23.1, 39.7) |

SD: standard deviation, NETCRF: Non-exercise testing cardiorespiratory fitness, METS: metabolic equivalent values.

[†]Recorded at birth or at 7y if missing at birth

*derived from the Clinical Interview Schedule-revised (CIS-R); [α]summarised as Median (25th,75th centile).

determined as 100×fraction missing information [31]. Initial investigations demonstrated that the complete case and imputed analyses were broadly similar; the latter are shown in the results.

## 4-way decomposition analysis

To estimate the relative contribution of the potential pathways linking mid-adult PA to subsequent cognitive function, we used a four-way decomposition analysis. This methodology allows us to assess the extent to which the estimated overall effect ($_e$OE) of PA on cognitive function is explained by alternative pathways involving (or not involving) NETCRF. Specifically, the method decomposes the estimated overall PA—cognitive function effect into 4 sub-components: an estimated controlled direct effect ($_e$CDE); an estimated randomised analogue of the pure indirect effect ($_{er}$PNIE); an estimated randomised analogue of the reference interaction ($_{er}$INT$_{REF}$) and an estimated randomised analogue of the mediated interaction ($_{er}$INT$_{MED}$). Briefly, $_e$CDE is the portion of the $_e$OE of PA on cognitive function due to pathways that do not involve NETCRF (neither mediation nor interaction). $_{er}$PNIE is the estimated effect only due to the mediation pathway that does not involve interaction. $_{er}$INT$_{REF}$ is the estimated effect of PA on cognitive function which is due solely to the interaction between PA and NETCRF (i.e., effect of PA varies by fitness level). $_{er}$INT$_{MED}$ is the estimated effect of PA on cognitive function due to both the interaction between PA and NETCRF and the mediating effect of NETCRF.

The four sub-components sum up to the $_e$OE of PA on cognitive function. In addition to the four-way decomposition of the $_e$OE described above, we also calculated randomised analogues for the proportions mediated by NETCRF, attributable to the PA-NETCRF interaction and the proportion eliminated (due to both mediation by, and interaction between NETCRF and PA). Notably, we account for intermediate confounders of the NETCRF—cognitive function association that are affected by PA (e.g., BMI), by estimating randomised analogue of effects. All effects were estimated on the difference scale, with 95% confidence intervals (CIs) calculated from bootstrapped standard errors (n = 100). See Fig 1 and S1 Table for further details.

## Supplementary analyses

To identify the extent to which unmeasured confounding may influence estimates, we used the E-value approach of Vanderweele & Ding [32]. An E-value is the minimum strength of association an unmeasured confounder would need to have with both PA and cognitive function, conditional on measured covariates, to fully explain away the PA–cognitive function association. Our estimated causal effects were on the difference scale; thus, they were transformed into risk ratios using the Vanderweele & Ding transfomation [32] before E-values were caluculated. We report the (i) E-value of the estimate and (ii) E-value for the limit of the CI closest to the null.

To determine whether effects varied by cognitive function domains, as previously suggested [12,13], we repeated the four-way decomposition analysis using each of the 4 individual cognitive function tasks as our outcome. Analyses were conducted in Stata MP v15.1 and R 4.3.0 (using the '*CMAverse*' [33] suite of functions).

## Results

### Sample characteristics

Approximately two thirds of males and females reported being physically active at least once/ week at 42y (Table 1). On average, NETCRF at 45y was slightly higher in males (11.8 METS

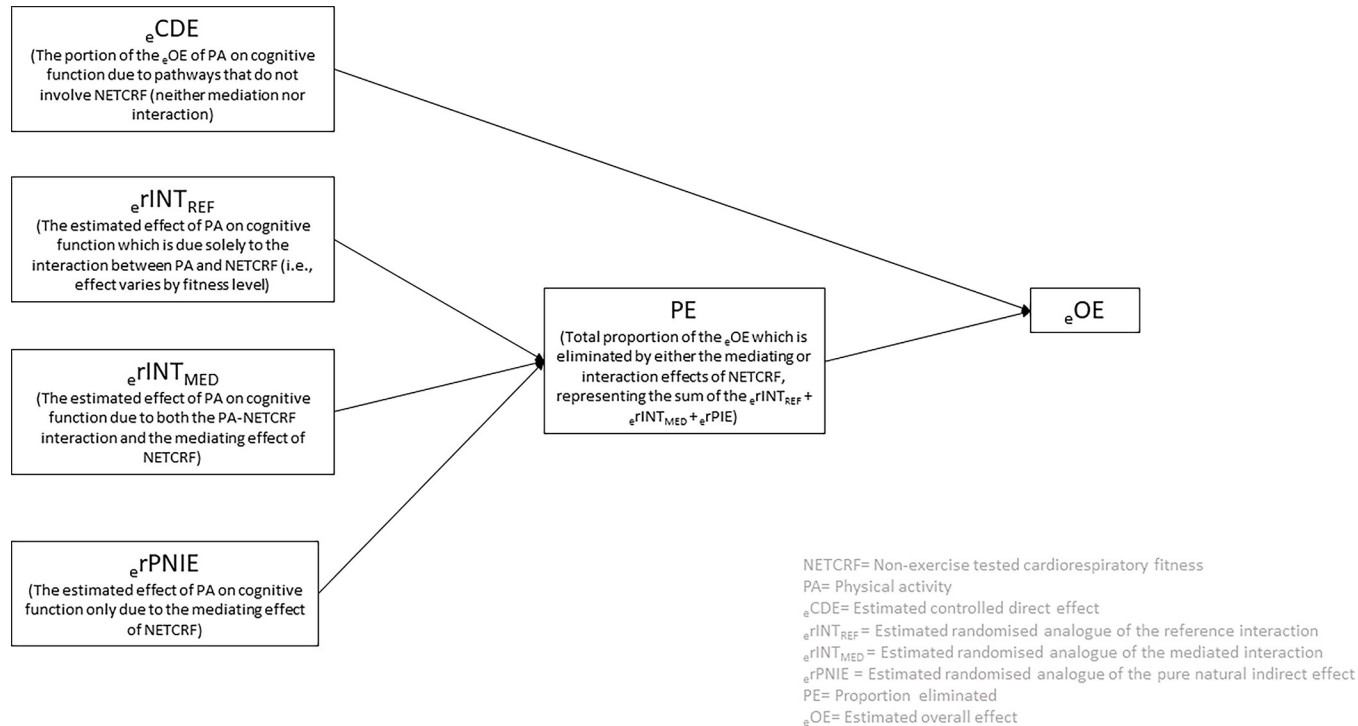

**Fig 1. Four-way decomposition of the estimated effect of physical activity frequency at 42y on cognitive function at 50y.**

(standard deviation (SD): 1.2) vs 9.0 METS (SD:1.3)), whereas females performed slightly better on all four cognitive tasks at 50y (S2 Table). Our analytical sample was broadly comparable to the original 1958 cohort participating in early life, with small differences in the analytical sample being more likely to be from a higher social class, have higher cognition scores at 11y and report higher levels of PA at 16y (S3 Table).

Association between 42y PA, 45y NETCRF and 50y overall cognitive function

42y PA was associated with higher 45y NETCRF; e.g., after adjusting for baseline confounders, males doing PA at least once/week had 0.44METS (95% CI: 0.36,0.53) higher NETCRF than those engaging less than once/week (Table 2). 42y PA was also associated with 50y overall cognitive function; e.g., in males, after adjusting for baseline confounders, PA at least once/week was associated with a 0.08 (95% CI: 0.04, 0.11) higher overall cognitive function z-score. Finally, while a 1-unit higher 45y NETCRF in both sexes was associated with higher 50y overall cognitive function in unadjusted models, associations were fully attenuated after adjustment for baseline confounders, 42y PA and intermediate confounders.

## 4-way decomposition analysis

**Males.** There was a small positive $_e$OE of 42y PA on 50y overall cognitive function (expressed as a difference in mean z-scores); i.e., the $_e$OE of PA at least once/week at 42y (vs. less than once/week) was a 0.08 (95% CI: 0.04,0.13) higher 50y overall cognitive function z-score. The $_e$CDE was similar to the $_e$OE (0.08 (0.03,0.12)), indicating that the PA—overall cognitive function association was mainly due to pathways excluding NETCRF. Thus, the $_e$rPNIE, $_e$rINTREF and $_e$rINTMED effects were all consistent with the null (Table 3).

**Table 2. Associations between Physical activity frequency, NETCRF and overall cognitive function (N = 9,385)\*.**

| | Mean difference in NETCRF (METS) (45y) (95% CI) | | Mean difference in overall cognition function z-score (50y) (95% CI) | | |
|---|---|---|---|---|---|
| | Model A | Model B | Model A | Model B | Model C\*\* |
| | | | Males | | |
| Physical activity frequency at 42y (ref: <once per week) | 0.52 (0.44, 0.61) | 0.44 (0.36, 0.53) | 0.15 (0.11, 0.20) | 0.08 (0.04, 0.11) | - |
| NETCRF at 45y (METS) | - | - | 0.04 (0.02, 0.06) | -0.00 (-0.02, 0.02) | 0.02 (-0.02, 0.06) |
| | | | Females | | |
| Physical activity frequency at 42y (ref: <once per week) | 0.59 (0.50, 0.68) | 0.51 (0.41, 0.60) | 0.11 (0.07, 0.15) | 0.03 (-0.01, 0.07) | - |
| NETCRF at 45y (METS) | | | 0.05 (0.04, 0.07) | 0.01 (-0.01, 0.03) | 0.03 (-0.01, 0.08) |

NETCRF: Non-exercise testing cardiorespiratory fitness; METS: metabolic equivalent values; CI: confidence interval.

\*based on n = 30 imputed datasets; Model A: unadjusted; Model B: adjusted for baseline confounders (socioeconomic position at birth, sports participation at 16y, cognitive function at 11y, academic attainment at 33y, smoking status at 42y, alcohol consumption at 42y); Model C: Model B + physical activity at 42y, BMI at 45y, physical activity at 45y, general health at 45y and sleep score at 45y

\*\*association between PA at 42y and total cognition z-score was not adjusted for NETCRF as this would be adjusting for the potential mediator/moderator which would attenuate the overall effect of physical activity which is the estimand of interest here.

## Females

The $_e$OE of PA at 42y on overall cognitive function at 50y was small and confidence intervals straddled the null (0.03 (95% CI: -0.01, 0.07)). While there was some evidence of mediation by NETCRF, all four estimated decomposed effects were consistent with the null (Table 3).

**Table 3. Estimated overall, controlled direct and randomised analogues of the pure indirect, reference interaction and mediated interaction effects\* of Physical activity frequency at 42y on overall cognitive function at 50y (mediated/moderated by NETCRF at 45y).**

| | Difference in mean overall cognitive function z-score (95% CI) | |
|---|---|---|
| | Physical activity (42y; ref: <once/week) | |
| | Males (n = 4,614) | Females (n = 4,771) |
| Overall effect ($_e$OE) | 0.08 (0.04, 0.13) | 0.03 (-0.01, 0.07) |
| Controlled Direct Effect ($_e$CDE) | 0.08 (0.03, 0.12) | 0.02 (-0.02, 0.07) |
| Randomized analogue of Pure Natural Indirect Effect ($_{er}$PNIE) | 0.01 (-0.01, 0.03) | 0.02 (-0.00, 0.04) |
| Randomized analogue of Reference Interaction ($_{er}$INTREF) | -0.00 (-0.00, 0.00) | -0.00 (-0.00, 0.00) |
| Randomized analogue of Mediated Interaction ($_{er}$INTMED) | -0.01 (-0.02, 0.01) | -0.01 (-0.02, 0.00) |
| Randomized analogue of overall proportion mediated ($_{er}$PM) | 0.05 (-0.11, 0.31) | 0.36 (-1.98, 2.68) |
| Randomized analogue of overall proportion attributable to interaction ($_{er}$INT) | -0.10 (-0.32, 0.12) | -0.27 (-1.71, 0.63) |
| Randomized analogue of overall proportion eliminated ($_{er}$PE) | 0.03 (-0.12, 0.31) | 0.33 (-1.75, 2.44) |

NETCRF: Non-exercise testing cardiorespiratory fitness; \*$_e$CDE is the portion of the $_e$OE of PA on cognitive function due to pathways that do not involve NETCRF (neither mediation nor interaction); $_{er}$PNIE is the effect only due to the mediation pathway that does not involve interaction; $_{er}$INTREF is the effect of PA on cognitive function which is due solely to the interaction between PA and NETCRF; $_{er}$INTMED is the effect of PA on cognitive function due to the effect of NETCRF as both a mediating and moderating variable. The four sub-components sum up to the $_e$OE of PA on cognitive function: see methods and supplementary materials for details.

### Supplementary analyses

The E-values for the causal estimates are presented in S4 Table. In males, the E-values suggest that the risk ratios for the $_eOE$ (RR = 1.11) and $_eCDE$ (RR = 1.11) could be fully explained by an unmeasured confounder that was associated with both PA at 42y and overall cognitive function at 50y by a risk ratio of approximately 1.47 and 1.46, respectively, independent of the measured confounders. The lower limit of the 95% CI could be moved to include the null by an unmeasured confounder that was associated with PA and overall cognitive function by a risk ratio of approximately 1.29 and 1.27, respectively. In females, smaller amounts of unmeasured confounding would be required to explain away the estimated effect of PA at 42y on overall cognitive function at 50y. Furthermore, because the confidence interval of the RR already straddled the null, unmeasured confounding was unneccessary to consider to move the CI to the null.

Associations reported above for overall cognitive function in males, was driven mainly by associations between PA and immediate and delayed verbal memory (S5–S8 Tables). For females, as per the main analysis, there was no estimated effect of PA on any of the four cognitive function sub-domains.

## Discussion

Using data from a large general population sample followed from birth for over five decades, we present novel findings, obtained from an under-utilised four-way decomposition approach, regarding the potential mediating and/or moderating effects of CRF on the relationship between PA frequency at 42y and overall cognitive function 8 years later. In males, we observed a small positive estimated overall effect of PA frequency on subsequent overall cognitive function, which was almost exclusively driven by an estimated direct effect of PA frequency, with no strong evidence in support of CRF either mediating or moderating the association. In females, once confounders were considered, there was no estimated overall effect of PA frequency on overall cognitive function.

While there is a general consensus that engagement in PA is beneficial for cognition [34], evidence of the benefits of PA engagement in mid-life is mixed, with some studies citing a lack of evidence [34], whilst others report beneficial effects [7]. In contrast to our findings, a recent study in 1,147 participants born in Britian in 1946, observed consistent positive associations between participation in lesisure time PA from 36y onwards, and cognitive function at 69y, though estimates were smallest and attenuated when looking only at PA engagement in early-to-mid-adulthood (36-43y) [35]. As different PA and cognitive function measures, assessed at a later life-stage (69y vs 50y), were used, drawing comparisons between the studies is difficult. However, as was observed in our study, it may be that PA only results in small benefits to subsequent cognitive function and thus sustained PA over a longer time period may be required in order accumulate these small incremental benefits that translate into tangible improvements in cognitive function. Alternatively, similar to associations between physical activity and depressive symptoms [36], the magnitude of the physical activity cognition association may vary by life-stage.

We found evidence that the PA−cognition relationship may be sex-specific, observing in males, a small positive estimated effect of PA at 42y and overall cognitive function 8 years later, but no effect in females. While we did observe an association between PA at 42y in females and subsequent cognitive function in unadjusted models, this association was attenuated to the null upon adjustment for baseline confounders, suggesting a potential important confounding pathway in females. This is in contrast to a recent report from the English Longitudinal Study of Ageing, which found evidence for a relationship between PA and cognitive

decline in females, but not in males [37]. Furthermore, evidence from randomised trials (which are less likely to be affected by confounding variables) in older participants (average age >70y) with a cognitive impairment diagnosis, also suggests that the relationship between PA and cognitive function is greater in females [28,38]. Discrepancies between findings could be due to differences between the life-stage examined or the measure of physical activity used (for example, the dichotomy of activity examined here, could have different implications by sex). Nonetheless, several mechanisms have been proposed to explain possible sex-specific effects of PA engagment on later cognition, including differences in: neuroplasticity, brain-derived neurotrophic factor (BDNF) levels and physiological adaptations to exercise [28].

Our work is novel because, to our knowledge, no other study has examined simultaneously the mediating and moderating role of CRF on the relationship between mid-life PA and subsequent cognitive function. Our formal mediation analysis is in line with a 2015 Cochrane review [39], which found no evidence that PA-induced increases in CRF resulted in improved cognitive performance in healthy older adults without known cognitive impairment. More recently, studies in children have observed a mediating effect of fitness on the relationship between PA and executive function [40] and academic attainment [41]. However, these studies, in children aged 9-13y, may not be generalisable to adults. Studies in adults report mixed findings, with some studies finding evidence of a mediating effect of fitness [42], whilst others have not [43]. If positive effects of mid-life PA on cognitive function are not due to increases in fitness, then what may be the mode of action? Several mechanisms have been proposed and include a reduced likelihood of vascular diseases and improvements in cerebral perfusion [44–46], stimulation of growth factors (e.g., brain-derived neurotrophic factor [47]), and/or the downregulation of oxidative stress and inflammatory responses [48]. Alternatively, PA may positively affect cognitive function, independent of the movement component of exercise, due to the cognitively stimulating aspects of PA, such as eye-hand coordination, visuospatial memory, self-motivation [49], planning and social interaction [50,51].

In trying to explain the apparent lack of a *PA-CRF-cognitive outcome* mechanism of action, Young et al (2015) suggested that PA effects may operate in certain subgroups of the population only, such as those starting from a lower baseline of CRF, i.e., a moderating effect may exist between PA and CRF [39]. A major novelty of our study, due to the use of the 4-way decomposition approach [15], is that we have been able to investigate the contribution that any PA-CRF interaction effect exerts on cognitive function. We observed no evidence of an interaction effect, such that in males, the positive effect of PA at 42y was consistent across the distributon of NETCRF at 45y. Previous studies have reported mixed findings regarding the interactive effect of fitness on the relationship between PA and cognitive outcomes, with some observing evidence of a greater benefit of PA in those with higher fitness [12], lower fitness [13], or no effect [52].

Our study has a number of strengths. For example, our use of a four-way decomposition analysis, has allowed us, for the first time, to more accurately model real world processes, thus avoiding potential biases inherent in simpler methods. We were able to account for intermediate confounders (e.g., BMI at 45y) which are themselves caused by our exposure (i.e., PA at 42y). Importantly, our methodology allowed us to robustly examine mediation and moderation simultaneously. This was possible due to the cohort's prospective design with multiple follow-up time points, enabling us to respect the temporal ordering of our exposure, mediator, outcome and confounding variables. Thus the possibility of reverse causation is reduced. Our outcome, overall cognitive function, was assessed at approximately the same age (50y) for all individuals, thus removing the known influence of age on cognitive function [53]. Our overall cognitive function variable was based on the average of four cognitive function tasks which have demonstrated validity in the prediction of incident dementia [54] and as a supplementary

analysis we examined each cognitive domain seperately. Nonetheless we acknowledge study limitations. For example, our approach relies on several assumptions including no unmeasured confounding of the PA-cognitive function, PA-NETCRF and NETCRF-cognitive function associations. While availability of detailed, prospectively collected covariate data enabled us to account for several important confounders, the possibility of residual confounding could not be ruled out. Therefore, we investigated the extent to which unmeasured confounding may be influencing our estimates, using E-values [32]. Moreover, our intermediate confounders (e.g., sleep problems) were measured at the same time as our mediator/moderator (NETCRF) and we acknowledge that the direction of association between factors measured contemporaneously are not always straight-forward to disentangle (in particular for sleep problems) [55–57]. Importantly, our binary exposure, PA at 42y, was based on self-report and limited to leisure-time activity, rather than total activity, which would include other domains (e.g., occupation). It may be subject to measurement error and bias and, due to datalimitations, we were unable to consider duration and intensity of PA at 42y. Nonetheless our 42y PA measure and analysis focusing on a counterfactual reality, whereby the least active group in the population (active less than once per week) become active (at least once per week; regardless of whether or not they meet recommended guidelines) has value, as supported by dose-response associations with mortality showing greatest benefit at the lower end of the physical activity spectrum (i.e., going from 'nothing to 'anything' [58]). Additionally, previous findings of our activity measure (e.g. with blood pressure) [59] provides construct validity and elsewhere has been associated with important health outcomes including mortality [60,61]. Our mediator, NETCRF at 45y, was predicted using an equation that has been well validated in adults [21]. While it is an appropriate proxy for CRF in large scale studies, such as the 1958 birth cohort, we acknowledge that NETCRF is not equivalent to the gold-standard assessment of CRF via tests to exhaustion measuring oxygen uptake [10]. Furthermore, our average NETCRF estimates indicate a relatively fit sample when compared to adults of a similar age in the general population [62]. While the cognitive function measures used have been adopted in other large-scale epidemiological studies [25,63] and have been shown to be predictive of incident dementia [26], they only reflect fluid cognition. As such, using a measure of crystallised cognition may have resulted in different associations being observed. As in all longitudinal studies, loss to follow-up occurred and while participants in mid-adulthood were broadly representative of the original population, the most disadvantaged were least likely to remain [17]. While statistical approaches such as inverse probability weighting and/or multiple imputation (the latter of which was adopted in this manuscript), could address potential bias due to non-response, attrition and sample selection [64], efforts should also be made to re-engage all cohort members with the study [17]. Finally, the 1958 birth cohort is predominantly of White British ethnicity (~98% at 45y). For the above reasons, generalisability (e.g., to other ethnic groups, less fit middle-aged men and women) should be inferred with caution.

We provide the first evidence obtained regarding the simultaneously mediating and moderating effect of CRF on the relationship between mid-life PA and subsequent cognitive function. We observed a small estimated total and direct effect of PA frequency at 42y on overall cognitive function at 50y in males only. In both sexes, there was no mediation or moderation of PA operating via fitness, suggesting that any positive effect of PA on cognitive function operates through other mechanisms which need to be explored.

## Supporting information

**S1 Fig. Sample flow diagram.**
(DOCX)

**S2 Fig. Directed acyclic graph (DAG)\*.**
(DOCX)

**S1 Table. Definitions of causal estimators for a continuous outcome (overall cognition z-score) with a binary exposure (PA at 42y) and continuous mediator (NETCRF at 45y).**
(DOCX)

**S2 Table. Scores on individual cognitive function tasks at 50y.**
(DOCX)

**S3 Table. Sample characteristics of original\* vs included sample (n = 9,385), at birth and in early life.**
(DOCX)

**S4 Table. E-values for estimated overall and controlled direct effects (expressed on risk ratio scale).**
(DOCX)

**S5 Table. Estimated overall, controlled direct and randomised analogues of the pure natural indirect, mediated and interaction effects of physical activity frequency at 42y on immediate verbal memory at 50y (mediated/moderated by NETCRF at 45y).**
(DOCX)

**S6 Table. Estimated overall, controlled direct and randomised analogues of the pure natural indirect, mediated and interaction effects of physical activity frequency at 42y on verbal fluency at 50y (mediated/moderated by NETCRF at 45y).**
(DOCX)

**S7 Table. Estimated overall, controlled direct and randomised analogues of the pure natural indirect, mediated and interaction effects of physical activity frequency at 42y on visual processing speed at 50y (mediated/moderated by NETCRF at 45y).**
(DOCX)

**S8 Table. Estimated overall, controlled direct and randomised analogues of the pure natural indirect, mediated and interaction effects of physical activity frequency at 42y on delayed verbal memory at 50y (mediated/moderated by NETCRF at 45y).**
(DOCX)

**S1 Text. Variable derivation.**
(DOCX)

## Acknowledgments

The authors are grateful to the Centre for Longitudinal Studies (CLS), UCL Institute of Education, for the use of the 1958 cohort data and to the UK Data Service for making them available. However, neither CLS nor the UK Data Service bear any responsibility for the analysis or interpretation of these data.

## Author Contributions

**Conceptualization:** Tom Norris, Snehal M. Pinto Pereira.

**Formal analysis:** Tom Norris.

**Investigation:** Tom Norris, Joanna M. Blodgett, Mark Hamer, Snehal M. Pinto Pereira.

**Methodology:** Tom Norris.

**Software:** John J. Mitchell.

**Supervision:** Snehal M. Pinto Pereira.

**Writing – original draft:** Tom Norris, Snehal M. Pinto Pereira.

**Writing – review & editing:** John J. Mitchell, Joanna M. Blodgett, Mark Hamer.

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
