## [Decision Letter · Decision Letter 0]

3 Apr 2024

PONE-D-23-37261Does cardiorespiratory fitness mediate or moderate the association between mid-life physical activity and cognitive function? Findings from the 1958 British birth cohort studyPLOS ONE

Dear Dr. Pinto Pereira,

Thank you for submitting your manuscript to PLOS ONE. After careful consideration, we feel that it has merit but does not fully meet PLOS ONE’s publication criteria as it currently stands. Therefore, we invite you to submit a revised version of the manuscript that addresses the points raised during the review process.

In particular, the reviewers identified a number of areas where the manuscript could be improved, particularly in relation to the underlying hypothesis between physical activity and cognition, use of self-reported measures and lack of a dose-response relationship.

We look forward to receiving your revised manuscript.

Kind regards,

Kathleen Bennett

Academic Editor

PLOS ONE

Journal Requirements:

- https://doi.org/10.1111/sms.14525

In your revision ensure you cite all your sources (including your own works), and quote or rephrase any duplicated text outside the methods section. Further consideration is dependent on these concerns being addressed.

"This work was funded by a UK Medical Research Council Career Development Award (ref: MR/P020372/1) awarded to SPP. JB is supported by a British Heart Foundation grant (SP/F/20/150002). JJM is funded by an MRC grant (MR/N013867/1). "

5. Please provide a complete Data Availability Statement in the submission form, ensuring you include all necessary access information or a reason for why you are unable to make your data freely accessible. If your research concerns only data provided within your submission, please write "All data are in the manuscript and/or supporting information files" as your Data Availability Statement.

Reviewers' comments:

Reviewer's Responses to Questions

**Comments to the Author**

1. Is the manuscript technically sound, and do the data support the conclusions?

Reviewer #1: Yes

Reviewer #2: Partly

2. Has the statistical analysis been performed appropriately and rigorously? 

Reviewer #1: Yes

Reviewer #2: I Don't Know

3. Have the authors made all data underlying the findings in their manuscript fully available?

Reviewer #1: Yes

Reviewer #2: Yes

4. Is the manuscript presented in an intelligible fashion and written in standard English?

Reviewer #1: Yes

Reviewer #2: Yes

5. Review Comments to the Author

Reviewer #1: Thank you for the opportunity to review this interesting, novel and potentially important study investigating the potential mediating and moderating effect of fitness in the association between physical activity and cognitive function in middle-aged adults. The link between physical activity and cognitive function is not clear-cut and studies investigating potential pathways are essential. The study includes a relatively large (n=9,385) sample and uses a novel approach to investigate mediation and/or moderation through fitness. The statistical methods used in the study are appropriate and are well explained in the text. The authors have used appropriate measures to account for missingness and potential confounding, as these are common issues in epidemiological research. In the results and the discussion, the findings are clearly presented, and potential explanations are well explored. I have only some minor comments that I hope the authors will consider in a potential revision:

Abstract

- Please present the proportion of women in the study. This is highly relevant as the authors present sex-specific findings.

Introduction

- The sentence “In the United Kingdom (UK), there are no effective treatments to reverse or delay dementia progression” makes it sound as if there are effective treatments for dementia in other countries. The lack of effective treatments for dementia is a worldwide issue, and I think this sentence understates how crucial it is to look elsewhere (modifiable risk factors). I would suggest rephrasing the sentence, either to just state that there are no effective treatments anywhere, or clarify otherwise.

- The authors state: “Additionally, CRF may also moderate the PA―cognition association, such that associations differ depending on fitness levels as higher fitness would enable participation in greater volumes/intensity of PA.” To me, the last part of the sentence sounds more like mediation (i.e. CRF -> PA -> cognitive function) rather than moderation (the association of PA with cognitive function differing by level of CRF). In my opinion, this sentence needs clarification/rephrasing.

Methods

- Outcome: please include the (average) time between the initial immediate verbal memory test and the delated verbal memory test.

- Statistical analysis:

o Some of the variables, including important exposure variables, had a quite high missing proportion. The authors have performed multiple imputation, but it could also be informative if the authors performed complete case analyses (sensitivity) so that the reader can better understand the potential bias caused by the high missingness.

o Have the authors considered using inverse probability weighting to account for non-response and potential bias due to skewed participation (e.g. related to education)?

Results

- Please include explanations of abbreviations in the tables.

- The authors present a list of potential confounders (also illustrated in the DAG in the supplementary) in the methods section. However, several of these potential confounders are not listed in Table 2 (model B or C). For instance, BMI and sleep problems are missing. Is this a typo, or where these not included in the models? Could the authors perhaps clarify in the methods section which potential confounders where included in the models (and why e.g. BMI and sleep problems were not included, if this is the case).

Supplementary

- DAG: I am not entirely convinced when it comes to the direction of the association between sleep problems at 45 years of age and CRF at the same age. It has been suggested that better CRF may improve sleep (see, for instance: Ernstsen et al., 2023 DOI: 10.1016/j.mayocp.2022.08.013; Dishman et al., 2015 DOI: 10.1249/MSS.0000000000000506), making sleep quality a potential mediator in the CRF-cognitive function association, rather than a confounder. Although it is unlikely that adjustment for sleep problems was the reason for the lack of association between NETCRF and cognitive function in this study, I think the authors should reconsider the inclusion of sleep problems as a confounder.

Reviewer #2: I enjoyed reading this submission, and really looked forward to the mediation/moderation analysis. On accepting the review, I very much was looking forward to how fitness was going to be justified in the background, as I admit, I don’t see why a relationship should exist. From this perspective, I was disappointed to find that little justification for looking at this relationship was offered. Given the findings were consistent with my own biases, that is no mediation/moderation effect, I can see how this could find a place in the literature supporting many others' views. But I do not feel easy with how this paper could be accepted in the literature, and I wonder if it is the way the data and analysis were used, I don’t believe there was a chance for any existing effect to be found. And I do not believe there is much recognition for these limitations. I am not sure what data you have available within this data set, but my concerns are around the self-report nature of PA, and the comparison of the existence of PA within the self-report data (at least once per week), or not (no record per week). Surely if fitness was to have a mediating or moderating effect, there would need to be a consideration of a dose-response, rather than the dichotomised approach of PA being present or not. These general mechanistic considerations seem much more important than a particular analytical approach, and so my major concerns with this manuscript centre there. I do however think that showing the existence of PA impacting later-life cognition is more important than recognised in the manuscript. I have made further specific comments as I read through the manuscript. Whilst the broad statistical approach makes sense to me, I cannot offer commentary on the specifics.

Specific comments:

Background:

Can there be a clearer justification of why CRF may have a mediating/moderating effect? I don’t really see why this should exist. There is nothing I can think of that could suggest this, apart from maybe a link to vascular health.. but I feel that is a stretch. A stronger argument may be saying that others have suggested it (if they have?) and so we sort to test it. Mechanistically though (biologically, not statistically), I struggle to see how the relationship should exist in that way, whereas the reverse argument is much clearer and obvious. The moderating argument also suggests that CRF may enable greater PA… but if this was the case, then analysing data that extends beyond a dichotomy of at least once per week, or less than once per week is surely warranted. So given the justification, it does not feel like the methods follow through with the current methodology.

Methods:

I am not sure I agree that there is strong validity of the prediction equations from the Jurca et al 2005 paper. It would be better to report the results they actually got. Could you also highlight differences in how the variables used in the prediction equations between those used in the Jucra paper, and in this manuscript? From what I can see, it does not sound like the self-report measures were similar, nor if measures such as resting heart rate were similarly assessed.

Why was the exposure (PA frequency) dichotomised? What is the rationale for this? It seems more detail is available. What issues are there in the analysis if the same measures (self-report PA) are used in both the PA and NETCRF? Can they truly be used to evaluate if one mediates/moderates another in the prediction of a third (independent) measure?

How sensitive do we think these cognitive measures are? Could that be the reason why a relationship is not observed? Equally, how sensitive do we think the self-report PA is? These do not appear to garner much attention in the discussion.

BMI (yrs) also appears to be used both within the creation of the mediator variable, and then appears as a potential confounder as well. Same with PA level (45 yrs), are these variables included multiple times in the analysis?

Can you clarify what missing data was imputed? And how this was done? It is not clear to me how some of it was calculated (e.g., BMI, PA) given some of this data is already in the imputation model.

Table 1: sports participation – I don’t think that is mentioned earlier in the document. Is this used as a potential confounder?

Discussion

In relation to the second paragraph, could it be that PA becomes more important as we get older, rather than accumulation of PA per se?

Third paragraph – given the dichotomy of the analysis – is this really a fair conclusion? That the impact is sex specific? Could it just be that the dose comparison between the two groups is different? But this is ignored in the current analysis?

Strengths – “to more accurately reflect likely processes in the real world”… given my comments earlier, particularly in relation to the likely dose effects of PA, I don’t see how this claim can be made. I don’t disagree it may be a better way of statistically understanding relationships, but without it being based on a strong rationale, this statement seems grossly inappropriate.

References

A few inconsistencies in the referencing… e.g., use of capitals etc.

6. PLOS authors have the option to publish the peer review history of their article (what does this mean?). If published, this will include your full peer review and any attached files.

Reviewer #1: **Yes: **Ekaterina Zotcheva

Reviewer #2: No

---

## [Author Response · Author response to Decision Letter 0]

8 May 2024

Please see attached response to reviewer document.

---

## [Editor Report · Decision Letter 1]

17 May 2024

PONE-D-23-37261R1Does cardiorespiratory fitness mediate or moderate the association between mid-life physical activity frequency and cognitive function? Findings from the 1958 British birth cohort studyPLOS ONE

Dear Dr. Pinto Pereira,

Thank you for submitting your manuscript to PLOS ONE. After careful consideration, we feel that it has merit but does not fully meet PLOS ONE’s publication criteria as it currently stands. Therefore, we invite you to submit a revised version of the manuscript that addresses the points raised during the review process. Most of the reviewers comments have been addressed in the response and the manuscript updated. However, there are a small number of minor comments that require further clarification below.

We look forward to receiving your revised manuscript.

Kind regards,

Kathleen Bennett

Academic Editor

PLOS ONE

Journal Requirements:

Additional Editor Comments:

The authors have addressed most of the comments from the two reviewers. There are a couple of additional comments that may need further clarification /updates.

1. In the response to reviewer 1 the authors refer to 'inverse probability weighting' being adopted in the manuscript - however, IPW is not explicitly stated anywhere in the manuscript. If it is included in some aspects of the analysis perhaps state this.

2. Reviewer 1 suggested complete case analysis may inform interpretation in respect of any bias and the authors have completed an analysis using only complete cases. The findings from this suggest that they differ very little from the imputed data which is reassuring. Perhaps inclusion of a sentence to refer to this in the manuscript might be informative for readers.

3. when referring to the supplementary table 3 - the text refers to 'analytical' and the table 'included' sample in comparison the original sample - perhaps for consistency the 'analytical' and 'included' are referred to in the same way.

4. Some of the references are not complete e.g. reference 64 is missing 'Research Square' before 2023 (this is a preprint) and to check other references for completeness

---

## [Editor Report · Decision Letter 2]

24 May 2024

Does cardiorespiratory fitness mediate or moderate the association between mid-life physical activity frequency and cognitive function? Findings from the 1958 British birth cohort study

PONE-D-23-37261R2

Dear Dr. Pinto Pereira,

We’re pleased to inform you that your manuscript has been judged scientifically suitable for publication and will be formally accepted for publication once it meets all outstanding technical requirements.

Kind regards,

Kathleen Bennett

Academic Editor

PLOS ONE
---

## [Editor Report · Acceptance letter]

29 May 2024

PONE-D-23-37261R2 

PLOS ONE

Dear Dr. Pinto Pereira, 

I'm pleased to inform you that your manuscript has been deemed suitable for publication in PLOS ONE. Congratulations! Your manuscript is now being handed over to our production team.

Kind regards, 

on behalf of

Professor Kathleen Bennett 

Academic Editor

PLOS ONE